# D-Artemis: A Deliberative Cognitive Framework for Mobile GUI Multi-Agents

## Abstract

Graphical User Interface (GUI) agents aim to automate a wide spectrum of human tasks by emulating user interaction. Despite rapid advancements, current approaches are hindered by several critical challenges: data bottleneck in end-to-end training, high cost of delayed error detection, and risk of contradictory guidance. Inspired by the human cognitive loop of Thinking, Alignment, and Reflection, we present D-Artemis—a novel deliberative framework in this paper. D-Artemis leverages a fine-grained, app-specific tip retrieval mechanism to inform its decision-making process. It also employs a proactive Pre-execution Alignment stage, where Thought-Action Consistency (TAC) Check module and Action Correction Agent (ACA) work in concert to mitigate the risk of execution failures. A post-execution Status Reflection Agent (SRA) completes the cognitive loop, enabling strategic learning from experience. Crucially, D-Artemis enhances the capabilities of general-purpose Multimodal large language models (MLLMs) for GUI tasks without the need for training on complex trajectory datasets, demonstrating strong generalization. D-Artemis establishes new state-of-the-art (SOTA) results across both major benchmarks, achieving a 75.8% success rate on AndroidWorld and 96.8% on ScreenSpot-V2. Extensive ablation studies further demonstrate the significant contribution of each component to the framework.

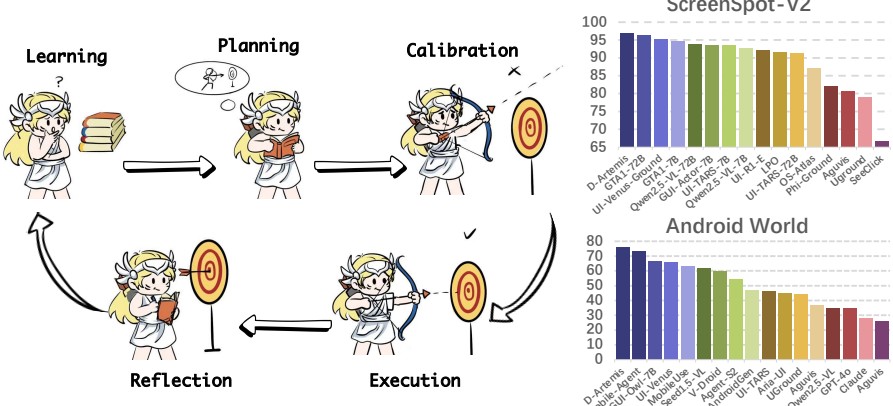

Figure 1: D-Artemis framework emulates the human cognitive loop of learning, planning, calibration, and reflection.

## 1 Introduction

Graphical User Interfaces (GUIs) Agents (Lai et al., 2025a; Bai et al., 2025; Wu et al., 2025b; Xie et al., 2025a; Liu et al., 2024; Dai et al., 2025) are designed to automate daily and professional tasks on various devices by emulating human-like interaction. Driven by the flexibility and versatility of mobile devices, the field of Mobile GUI agents (Agashe et al., 2025a; Li et al., 2025a; Gu et al., 2025; Ye et al., 2025; Qin et al., 2025) has witnessed rapid advancements in recent years.

Most early GUI agents leverage the accessibility (a11y) trees to identify UI elements (Rawles et al., 2024; Wanyan et al., 2025; Lai et al., 2025b; Xie et al., 2025b). To better emulate human-like perception and enhance agent robustness, recent research pivots towards vision-based agents that perceive the GUI directly from pixels. Current research on vision-based GUI agents largely follows two strategic directions. The first approach involves engineering agentic frameworks to augment the cognitive abilities of agents, for instance by enhancing reasoning with historical context or by improving state awareness through reflective processes (Agashe et al., 2025a;c; Li et al., 2025a). The second approach seeks to directly enhance the end-to-end capabilities of the core model through specialized training paradigms. This often involves techniques such as reinforcement learning (RL) on GUI trajectories to optimize decision-making, or fine-tuning on curated datasets designed to bolster the fundamental skills of the model, like visual grounding or self-verification abilities (Gou et al., 2025; Ye et al., 2025; Qin et al., 2025; Gu et al., 2025).

Despite this remarkable progress, significant challenges remain. 1) **Data Bottleneck in End-to-End Training.** While end-to-end training methods often rely on the automated generation of mobile GUI trajectory data to bypass the high costs of manual labeling, they are fundamentally constrained by the limited scope of their data sources. This constraint compromises the diversity of the resulting datasets, leading to models with diminished instruction-following abilities and poor compatibility across GUIs developed with different frameworks. 2) **High Cost of Delayed Error Detection.** Most framework methods employ the post-execution reflection strategy (Agashe et al., 2025a; Li et al., 2025a), meaning errors are detected only after a flawed action has already derailed the task trajectory. Furthermore, the feedback from this reflection is often limited to a conclusive judgment (i.e., success or failure), lacking the diagnostic information necessary for the agent to refine its faulty logic. Consequently, the agent not only incurs significant overhead for error recovery but is also prone to getting trapped in a vicious cycle of repeated failures. 3) **Risk of Contradictory Guidance.** A common strategy in agentic frameworks is to provide a generic set of tips guidance or example trajectories as an external knowledge source to bolster agent performance (Xie et al., 2025b; Agashe et al., 2025b; Lai et al., 2025a). However, in GUI tasks, even similar objectives often require different operational logic across applications. Potentially contradictory information can therefore introduce conflicting guidance, paradoxically hindering rather than helping the decision-making.

Human cognition in complex tasks often follows a deliberative cycle of learning, planning, calibration, and reflection (FIgure 1). Inspired by this cognitive model, we introduce the D-Artemis framework, which instantiates this process for GUI agents through a core workflow of thinking, alignment, and reflection. By emulating this human-like cognitive loop,it achieves significantly more robust and adaptive autonomous operation. D-Artemis employs a fine-grained, app-specific tip retrieval mechanism to provide highly relevant tips guidance, which avoids the logical conflicts of coarse-grained methods and enhances the effectiveness of guidance. Crucially, D-Artemis features a proactive Pre-execution Alignment stage, where lightweight Thought-Action Consistency(TAC) Check module and Action Correction Agent (ACA) cooperate to check and correct actions before execution, emulating human-like deliberation to prevent costly trajectory deviations. To complete the cognitive loop, a Status Reflection Agent (SRA) performs a post-execution strategic reflection, assessing the effectiveness of each step and generating insights to inform future decisions. A key advantage of D-Artemis is that it enhances the performance of general-purpose Multimodal large language models (MLLMs) on GUI tasks without training on complex trajectory datasets, demonstrating remarkable generalization capabilities. In summary, our contributions are four-fold:

- We introduce D-Artemis, a new vision-based agentic framework that integrates fine-grained tip guidance, proactive pre-execution alignment, and strategic post-execution reflection to effectively execute complex mobile GUI tasks.
- We propose a effective tip guidance mechanism that conducts fine-grained, app-specific retrieval, avoiding the logical conflicts caused by heterogeneous tips for similar tasks across different applications.
- We design a deliberative loop that combines pre-execution alignment with post-execution reflection, empowering the agent to emulate human process of fine-tuning actions and reflectively learning from outcomes.
- Extensive experiments on the AndroidWorld and ScreenSpot-V2 benchmarks demonstrate that D-Artemis achieves state-of-the-art (SOTA) performance in GUI automation and exhibits strong generalization capabilities. Moreover, thorough ablation studies confirm the significant contribution of each proposed components.

## 2 RELATED WORK

**GUI Agents.** Early GUI agents relied on structured data like Accessibility Trees (Rawles et al., 2024; Wanyan et al., 2025; Lai et al., 2025b; Xie et al., 2025b), but high costs and noise issues have spurred a shift toward vision-based approaches, leading to two mainstream architectures (Cheng et al., 2024). Single-agent models aim to enhance a core capabilities through several strategies: large-scale pre-training (Hong et al., 2024; Qin et al., 2025; Guo et al., 2025), innovative data generation techniques (Wu et al., 2025b; Xu et al., 2025), and post-hoc refinement mechanisms (Gu et al., 2025; Ye et al., 2025). In contrast, multi-agent frameworks improve efficiency through modular cooperation (Wang et al., 2024; Ye et al., 2025; Dai et al., 2025), but their reliance on inefficient post-execution verification means they cannot prevent flawed actions from causing erroneous state transitions, severely impacting overall performance.

**Post-Training Paradigms for GUI Understanding.** To equip vision-based agents with GUI-specific capabilities, two post-training paradigms are prevalent. Supervised Fine-Tuning (SFT) has produced powerful grounding models (You et al., 2023; Cheng et al., 2024; Lu et al., 2024), increasingly leveraging novel synthetic data generation pipelines to address data acquisition challenges (Gou et al., 2025; Wu et al., 2025b; Xu et al., 2025). More recently, Reinforcement Learning (RL) has gained traction to overcome the limitations of static SFT data. Modern RL approaches mitigate earlier challenges with long-horizon reasoning through techniques such as test-time planning with judge mechanisms (Yang et al., 2025a) and dense reward or preference optimization (Gu et al., 2025; Tang et al., 2025). However, the heavy dependence of both SFT and RL paradigms on large-scale training data underscores the value of our framework, which can significantly boost the performance of general-purpose models on GUI tasks without such data-intensive training.

**Retrieval-Augmented Generation (RAG) for Agents.** To improve reasoning and provide agents with relevant knowledge at inference time, many works employ techniques from RAG (Fan et al., 2024). In the agent domain, this often involves augmenting the input prompt by retrieving task exemplars (Kim et al., 2024), state-aware guidelines (Fu et al., 2024), or past trajectories from a memory module (Kagaya et al., 2024). Our work departs from prior methods by employing a fine-grained, app-specific tip retrieval strategy. This allows us to reduce informational noise and avoid logical contradictions in the guidance, leading to a significant improvement in its effectiveness.

## 3 METHOD

D-Artemis, illustrated in Figure 2, is a novel framework designed for complex mobile GUI automation tasks. The framework operates on a three-stage lifecycle for each step: action generation, pre-execution alignment, and post-execution reflection. The combination of task-specific tips from the knowledge base and a continuously updated working memory equips the manager agent with two crucial capabilities: task-oriented adaptability and real-time state awareness. The Thought-Action Consistency Check (TAC) module serves as a pre-execution safeguard. It efficiently classifies whether a thought-action pair is consistent or not, enabling proactive prevention of a significant number of invalid operations. Action Correction Agent (ACA) is triggered to diagnose the action error and apply a tailored correction. This proactive, pre-execution correction serves as a crucial alignment mechanism. It not only enhances the likelihood of a successful execution but, more critically, mitigates the risk of derailing the entire task trajectory, which could be caused by a single flawed action. Post-execution, Status Reflection Agent (SRA) assesses the outcome and generates strategic guidance for the next step. This core learning loop enables the agent to learn from experience and avoid repeating mistakes.

### 3.1 ACTION GENERATION

The manager agent serves as the primary action generator, taking the user-provided task $T_u$ and the environment observation $O$ (screenshot only) as input. D-Artemis adopts a fine-grained, app-specific retrieval strategy. For the given task, it begins by querying the knowledge base $\mathcal{K}$ for a concise set of highly relevant tips, denoted as $P_{T_u}$. Recognizing that similar tasks often demand different operational logic in different applications, our knowledge base is designed around app-specific modules of tips. Further details are provided in the Appendix G. The retrieval is therefore targeted to the specific applications within the task $T_u$, which avoids the critical issue of conflicting

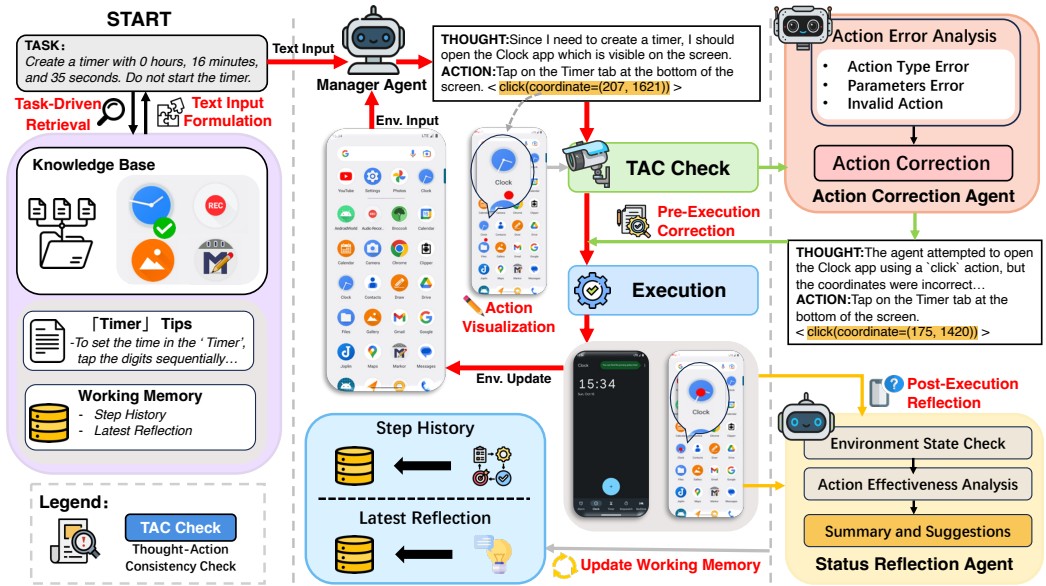

Figure 2: Overview of the D-Artemis framework. (a) The manager agent is guided by two input modalities: textual (task, tips, working memory) and visual (screenshot only). (b) Pre-execution, TAC Check module verifies thought-action consistency. (c) A low consistency score triggers the Action Correction Agent (ACA) to analyze the error type and rectify the action. (d) Post-execution, the Status Reflection Agent (SRA) assesses the action effectiveness and the environmental state to produce guidance for the next step. Upon completion of each step, the working memory is updated.

operational logic that arises from coarse-grained retrieval methods. The process is formally defined as:

$$P_{T_u} = \text{RetrieveTips}(\mathcal{K}, \text{App}(T_u))$$

The working memory $M_t$ is initialized at the onset of task, comprising two key components: step history and last reflection. The step history, denoted $H_t$, archives the thought-action pairs from previous steps. We define the record of a single step $t$ as $s_t = \langle \tau_t, a_t \rangle$. To maintain a focus on recent events, it is implemented as a sliding window with a size of five. This augmentation equips the agent with crucial insight into the intent of its past steps, which is pivotal for preventing action loops and for accurately tracking task progression. Furthermore, the latest reflection $R$ is continuously updated with the output $r$ generated by the SRA at the end of each step (discussed in Section 3.3).

$$M_t = \langle H_t, R_t \rangle = \langle (s_i)_{i=\max(1,t-5)}^{t-1}, r_{t-1} \rangle$$

Ultimately, the behavior of manager agent can be modeled as a policy $\pi$ that maps all available information to a thought-action pair. This process is formally expressed as:

$$s_t = \langle \tau_t, a_t \rangle = \pi(T_u, O_t, P_{T_u}, M_t)$$

## 3.2 PRE-EXECUTION ALIGNMENT

The pre-execution alignment mechanism is a cornerstone of the D-Artemis framework, designed to emulate the deliberate, fine-grained control that humans exhibit when interacting with mobile devices. In essence, humans do not operate in a purely reactive loop. After establishing an objective, they perform a crucial step of proactive calibration—adjusting their intended action (e.g., the precise tap location) to ensure it aligns perfectly with their goal before execution. This deliberative process stands in stark contrast to a simplistic "act-then-observe" cycle, where actions are executed without such prior validation. The nature of mobile GUI tasks is inherently sequential, with each action potentially altering the environmental state. Consequently, a single misstep can derail the entire trajectory, requiring a costly and extensive sequence of corrective actions to recover. Driven by its two core components— TAC Check and ACA—the pre-execution alignment mechanism significantly enhances both step-level efficiency and the overall fidelity of the task trajectory by proactively minimizing the likelihood of execution errors.

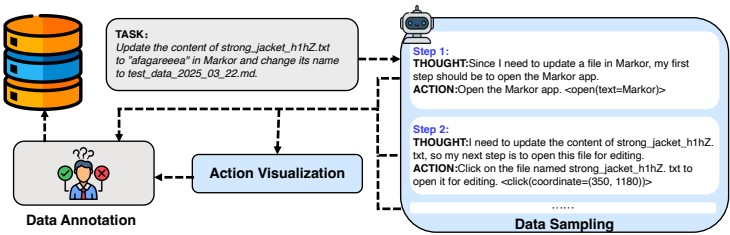

Figure 3: TAC data construction workflow.

### 3.2.1 THOUGHT-ACTION CONSISTENCY CHECK

To prevent the overhead of unnecessary corrections, we train a TAC check module. This lightweight expert model acts as an efficient filter, judging whether the proposed action $a_t$ aligns with the objective specified by the thought $\tau_t$. Its decision is based on two inputs: the thought $s_t$ and the action visual representation $V_{a_t}$, which is generated via action visualization.

$$c_t = \text{TAC}(\tau_t, a_t, V_{a_t}),$$

where $c_t \in \{0, 1\}$ denotes the output of TAC module. In the following, we discuss the design of data construction and training for this module.

**Data Sampling.**  As illustrated in Figure 3, to construct the training dataset, we first generated task execution trajectories in the AndroidWorld environment with the Qwen2.5-VL-72B-Instruct model  (Bai et al., 2025). As the function of TAC is to assess consistency at the step level, we then unrolled these trajectories, treating each thought-action pair as an individual data point. Following a subsequent cleaning and filtering stage, we curated a final dataset of 2,247 samples.

**Action Visualization.**  For each sample, we generate a visual representation of the proposed action ($V_{a_t}$) by annotating the corresponding screenshot. Recognizing that not all action types are readily visualizable, our approach specifically targets coordinate-based actions (e.g., *click*, *swipe*, *long press*). We employ distinct visual markers for each action type, rendered at their specified coordinates, to create an intuitive depiction of the intended operation. Further details are provided in the Appendix I.1. This multi-modal fusion provides the TAC module with a richer, more contextualized input, significantly enhancing its ability to comprehend the intent of action and perform accurate reasoning.

**Data Annotation.**  The TAC dataset was annotated by a dedicated team of six trained experts. To ensure consistency, the team held periodic calibration meetings to standardize the annotation criteria. Each data point underwent a multi-stage quality control process, including cross-reviews and a final audit by a senior expert. The inter-annotator agreement (IAA) achieved a Fleiss Kappa score (Fleiss, 1971) of 0.83, indicating almost perfect agreement. This rigorous process resulted in a high-fidelity dataset, which will be open-sourced to foster future research within the community. Further details regarding the annotation can be found in the Appendix F.

We utilized this dataset to fine-tune Qwen2.5-VL-7B (Bai et al., 2025) via SFT, creating our lightweight TAC module. Further details can be found in the Appendix D. The lightweight design is a key advantage, allowing the framework to cost-effectively prevent flawed actions through rapid pre-execution checks.

### 3.2.2 ACTION CORRECTION AGENT

Based on the extensive dataset collected in Section 3.2.1, we performed both a quantitative and qualitative analysis of the failure cases (the results are shown in Figure 4). Our analysis categorizes the errors into three primary types:

**Action Type Error.**  This error occurs when the type of the generated action type of $a_t$ does not match the intent of the thought $\tau_t$. For instance, the thought required *long press* to select text, but the executed action was *click*.

**Action Parameters Error.**  This was the most prevalent category of error. In these cases, the action type is correct, but its parameters are flawed. Common examples include incorrect coordinates for a *click* or the wrong text argument for a *type* action.

**Invalid Action.** In some scenarios, the model hallucinates and generates an action that falls outside the predefined action space.

If the TAC check fails ($c_t = 0$), ACA (denoted as $f_{AC}$) is triggered. It takes the complete thought-action context as input— comprising thought $\tau_t$, originally proposed action $a_t$, and its visualization $V_{a_t}$— to perform analysis. Its primary function is to first determine the category of error by matching the action against the predefined types and then apply a tailored rectification strategy. This process outputs a revised thought-action pair, $\langle \hat{\tau}_t, \hat{a}_t \rangle$. Notably, the visual input $V_{a_t}$ is indispensable here, as the spatial and semantic context it provides is instrumental in resolving the most prevalent Action Parameters Error.

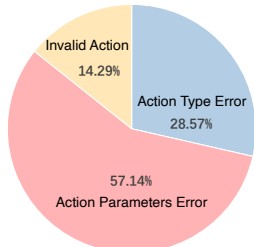

Figure 4: Distribution of Error Categories by Proportion.

$$s_t = \langle \hat{\tau}_t, \hat{a}_t \rangle = \begin{cases} f_{AC}(\tau_t, a_t, V_{a_t}), & \text{if } c_t = 0 \\ \langle \tau_t, a_t \rangle, & \text{if } c_t = 1 \end{cases}$$

### 3.3 POST-EXECUTION REFLECTION

The SRA is the core component responsible for the post-execution reflection process. Pre-execution alignment ensures thought-action consistency but cannot assess thought soundness. Higher-level reflection is thus crucial for overall task context. To perform this reflection, the SRA (denoted as $f_{SR}$) takes the overall task $T_u$, the executed thought-action pair $s_t$, and the environmental state transition $(O_t \rightarrow O_{t+1})$ as input. Its first function is to judge the step effectiveness by verifying if the outcome aligns with the objective of $\tau_t$. If the step is judged as a failure, the agent performs a deeper analysis: it summarizes the current situation and generates strategic guidance to avoid repeating the mistake. This entire output, denoted $r_t$, subsequently updates the last reflection within the working memory $M_{t+1}$, thereby informing the subsequent decision-making process:

$$r_t = f_{SR}(T_u, s_t, O_t, O_{t+1}).$$

## 4 EXPERIMENTS

### 4.1 EXPERIMENTAL SETTINGS

We evaluate the performance of D-Artemis on two key capabilities, dynamic task execution and GUI element grounding, using the widely-used **AndroidWorld** and **ScreenSpot-V2** benchmarks, respectively. The specific implementation details are as follows.

**AndroidWorld.** For dynamic task execution, we evaluate D-Artemis on AndroidWorld (Rawles et al., 2025), an online mobile agent benchmark that runs in a live Android emulator. It contains 116 core tasks across 20 applications. Through parameter randomization, these tasks yield millions of unique variants, testing a model's adaptability to diverse instructions and dynamic UI states.

**ScreenSpot-V2.** We evaluate the GUI element grounding performance of D-Artemis on ScreenSpot-V2 (Wu et al., 2025b). This is a general-purpose, cross-platform benchmark that measures a model's ability to localize UI elements in common scenarios. For our evaluation, we specifically utilize the mobile data subset of this benchmark. The dataset comprises 1,272 single-step instructions with corresponding bounding boxes for target elements, which include text-based elements, icons (e.g., the trash can icon), and widgets (e.g., to-do lists).

**Settings & Baselines.** We employ the open-source multimodal language model Qwen2.5-VL-72B-Instruct (Bai et al., 2025) and GUI-Owl-32B (Ye et al., 2025) as the base models, with the decoding temperature fixed at 0 to ensure deterministic outputs. All experiments were conducted on a server equipped with four $8 \times$ NVIDIA A100 80G GPUs. Detailed prompt templates are provided in the Appendix H. To demonstrate the effectiveness of our approach, we benchmark D-Artemis against a comprehensive suite of state-of-the-art (SOTA) methods. Further details on the experimental setup and the baselines can be found in Appendix C.

Table 1: Success Rate (%) on the AndroidWorld benchmark. The best score is highlighted in bold. The asterisk "†" indicates models trained on the GUI trajectory dataset.

| Category | Method | Model | SR↑ |
|---|---|---|---|
| Closed-source Models | Gemini (Team et al., 2024) | Gemini-1.5-Pro | 22.8 |
| | Claude (Anthropic, 2024) | Claude Computer-Use | 27.9 |
| | GPT-4o (Achiam et al., 2023) | GPT-4o | 34.5 |
| | Aguvis (Xu et al., 2025) | GPT-4o + Aguvis | 37.1 |
| | UGround (Gou et al., 2025) | GPT-4o | 44.0 |
| | Aria-UI (Yang et al., 2025b) | GPT-4o + Aria-UI | 44.8 |
| | AndroidGen (Lai et al., 2025b) | GPT-4o | 46.8 |
| | Agent-S2 (Agashe et al., 2025c) | Claude-3.7-Sonnet | 54.3 |
| General Open-source Models | Aguvis (Xu et al., 2025) | Qwen2-VL-72B-Instruct† | 26.1 |
| | Qwen2.5-VL (Bai et al., 2025) | Qwen2.5-VL-72B-Instruct | 35.0 |
| | UI-TARS (Qin et al., 2025) | Qwen2.5-VL-72B-Instruct† | 46.6 |
| | Seed1.5-VL (Guo et al., 2025) | Seed1.5-VL | 62.1 |
| | MobileUse (Li et al., 2025b) | Qwen2.5-VL-72B-Instruct | 62.9 |
| | UI-Venus (Gu et al., 2025) | Qwen2.5-VL-72B-Instruct† | 65.9 |
| GUI-specific Models | V-Droid (Dai et al., 2025) | V-Droid | 59.5 |
| | Mobile-Agent-v3 (Ye et al., 2025) | GUI-Owl-7B | 66.4 |
| | Mobile-Agent-v3 (Ye et al., 2025) | GUI-Owl-32B | 73.3 |
| Ours | **D-Artemis** | Qwen2.5-VL-72B-Instruct | 68.1 |
| | **D-Artemis** | GUI-Owl-32B | **75.8** |

Table 2: Success Rate (%) on the Mobile Subset of the ScreenSpot-V2 Benchmark. D-Artemis utilizes Qwen2.5-VL-72B as the backbone model.

| Category | Method | Text | Icon/Widget | Avg |
|---|---|---|---|---|
| Closed-source Models | GPT-4o (Achiam et al., 2023) | 26.6 | 24.2 | 25.6 |
| General Open-source Models | Qwen2.5-VL-7B (Bai et al., 2025) | 98.3 | 85.3 | 92.8 |
| | Qwen2.5-VL-72B (Bai et al., 2025) | 97.6 | 88.6 | 93.8 |
| GUI-specific Models (SFT) | SeeClick (Cheng et al., 2024) | 78.4 | 50.7 | 66.7 |
| | UGround (Gou et al., 2025) | 75.1 | 84.5 | 79.1 |
| | Aguvis (Xu et al., 2025) | 89.3 | 68.7 | 80.6 |
| | OS-Atlas (Wu et al., 2025b) | 95.2 | 75.8 | 87.0 |
| | UI-TARS-72B (Qin et al., 2025) | 94.8 | 86.3 | 91.2 |
| | UI-TARS-7B (Qin et al., 2025) | 96.9 | 89.1 | 93.6 |
| | GUI-Actor (Wu et al., 2025a) | 97.6 | 88.2 | 93.6 |
| GUI-specific Models (RL) | Phi-Ground (Zhang et al., 2025) | 96.5 | 62.0 | 82.0 |
| | UI-R1-E (Lu et al., 2025) | 98.2 | 83.9 | 92.2 |
| | LPO (Tang et al., 2025) | 97.9 | 82.9 | 91.6 |
| | GTA1-7B (Yang et al., 2025a) | 99.0 | 88.6 | 94.6 |
| | GTA1-72B (Yang et al., 2025a) | 99.3 | 92.4 | 96.4 |
| Ours | **D-Artemis** | **99.3** | **93.4** | **96.8** |

## 4.2 MAIN RESULTS

**AndroidWorld.** Table 1 presents the performance comparison between D-Artemis and baseline models. Our framework sets a new state-of-the-art (SOTA) with a 75.8% success rate, a 2.5% absolute improvement over the GUI-specific Mobile-Agent-v3. These results clearly demonstrate the superior performance of D-Artemis on mobile GUI automation tasks. Furthermore, within the cohort of methods also employing Qwen2.5-VL-72B-Instruct, D-Artemis also establishes the state-of-the-art at 68.1%, surpassing the strong UI-Venus baseline by 2.2%. This confirms that our novel deliberative cognitive framework can significantly boost the capabilities of general-purpose models for GUI tasks, independent of advantages from model scale or data.

**ScreenSpot-V2.** As detailed in Table 2, D-Artemis demonstrates exceptional performance on the ScreenSpot-V2 benchmark. It achieves a 97.9% average success rate, surpassing the previous SOTA, UI-Venus-Ground-72B. The ability of Pre-execution Alignment to effectively correct the grounding of UI elements is demonstrated by its performance on the more challenging "Icon/Widget" tasks, where it reaches 95.6%. Crucially, D-Artemis outperforms its own base model, Qwen2.5-VL-72B, by a significant 4.1%. Our analysis reveals that pre-execution alignment significantly improves UI element grounding in common scenarios by proactively correcting flawed actions, demonstrating the overall effectiveness of our framework.

### 4.3 ABLATION STUDY

To validate the contribution of each module in D-Artemis, we designed and conducted a comprehensive set of ablation studies on the AndroidWorld benchmark. For a fair comparison, Qwen2.5-VL-72B was used as both the baseline model and the foundational LLM backbone for all experimental variants of our framework.

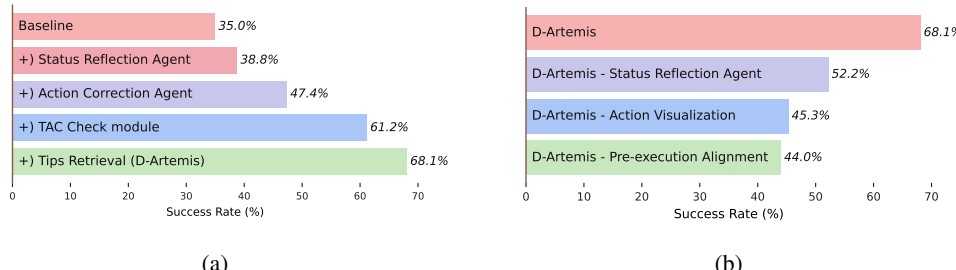

(a)            (b)

Figure 5: Ablation study on AndroidWorld.

**Pre-execution Alignment improves overall performance by enhancing step-level action effectiveness.** The Pre-execution Alignment process in D-Artemis involves the collaboration of TAC Check module and ACA. To assess the individual contribution of each component, we conducted a series of ablation studies where we systematically removed each module and observed the impact on task performance. The results are shown in Figure 5a. The ablation study demonstrates a tiered performance gain: adding the ACA alone improves the success rate by 8.6%, and further introducing the TAC Check module increases the total gain to 22.4%. The initial gain stems from improved execution efficiency, while the larger boost from the TAC module highlights its dual function as both an effective error filter and a safeguard against incorrect modifications. As shown in Figure 5b, removing the entire pre-execution alignment mechanism leads to a significant drop in performance, which highlights its importance to the framework. Furthermore, the significant performance drop observed upon removing action visualization demonstrates the visual information is crucial in the design.

**Post-execution Reflection enhances the state awareness of the agent.** The effectiveness of the SRA is quantified in Figure 5. On the baseline model, its inclusion yields a 3.8 % performance gain. This effect is significantly amplified within the D-Artemis framework, where the agent contributes 15.9% improvement. This highlights its dual capability to enhance perception of environmental changes and to provide effective guidance that informs the decision-making process.

**Tip Retrieval mechanism improves the decision-making ability of agent by minimizing conflicting guidance.** As detailed in Figure 5a, integrating the tip retrieval mechanism yields a significant 6.9% performance gain for D-Artemis. This improvement is particularly noteworthy because the base D-Artemis framework is already adept at accurately translating a given thought into its corresponding action. Therefore, this gain stems from the enhanced decision-making capabilities conferred by the task-specific tips. To further evaluate our app-specific tip retrieval strategy, we conduct a comparative study against two baseline strategies (as shown in Figure 6): (1) "without tips" baseline, which uses no external guidance, and (2) "mixed tips" baseline, which provides a generic, heterogeneous mixture of tips from different applications. Detailed information regarding the applications can be found in the Appendix 6. Our results reveal that providing the agent with a generic mixture of tips is often worse than providing no tips at all. This highlights a critical flaw in untargeted guidance: the introduction of noisy and logically conflicting information can paradoxi-

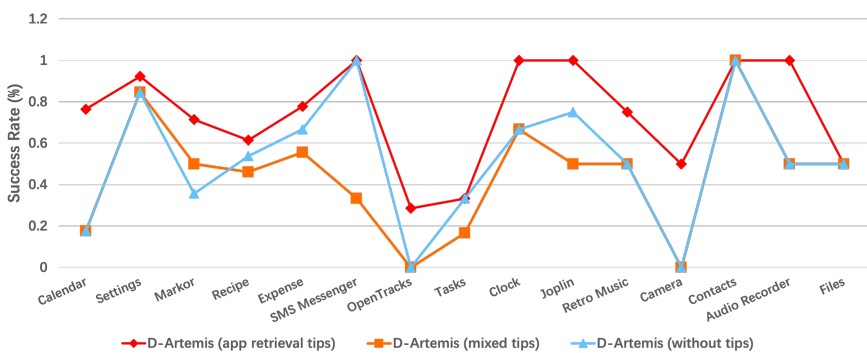

Figure 6: Success rates of different tip guidance strategies across AndroidWorld applications.

cally degrade, rather than improve, the decision-making process of the agent. This outcome, which is contrary to the very purpose of providing tips, validates the need for our tip retrieval strategy.

### 4.4 ERROR ANALYSIS

Table 3: The statistic of Error Rate(%) on AndroidWorld that D-Artemis failed to complete. Qwen2.5-VL-72B is used as both the baseline model and the foundational LLM backbone for D-Artemis. Method marked with "†" uses GUI-OWL-32B as backbone.

| Error Metric | Planning | Navigation | Grounding | Perception | Others |
|---|---|---|---|---|---|
| Baseline | 42.3 | 34.6 | 73.1 | 30.8 | 30.8 |
| D-Artemis | 75.0 | 12.5 | 5.0 | 62.5 | 20.0 |
| D-Artemis† | 35.7 | 10.7 | 3.6 | 71.4 | 14.3 |

We conducted a thorough error analysis of the failure cases for D-Artemis on the AndroidWorld benchmark. Inspired by the methodology of Li et al. (2025a), we categorized the errors into five distinct types. Detailed descriptions of each error category are provided in Appendix E.

For our error analysis, we examined the trajectories of failed tasks from each method. We then identified and classified each error according to the five predefined categories, noting that a single failed task can contain multiple errors. Finally, we calculated the proportional distribution of each error type relative to the total number of observed errors. The results of this analysis are presented in Table 3. Compared to the baseline, D-Artemis shows a substantial reduction in errors related to *Grounding* and *Navigation*, a result that directly reflects the architectural advantages of our framework. Notably, the majority of the remaining failures are now attributed to higher-level *Planning* and *Perception* errors. This suggests that while our framework effectively resolves issues of execution and guidance, the final performance is still bound by the inherent, end-to-end reasoning limitations of the underlying base model. A case study of error occurrence can be found in Appendix I.3.

## 5 CONCLUSION

In this work, we presented D-Artemis—a novel deliberative multi-agent framework designed to enhance the reliability and efficiency of mobile GUI agents by emulating a human-like cognitive process. Through the D-Artemis framework, we show the significant benefits of a proactive, deliberative control loop over traditional reactive, post-hoc reflection methods. By leveraging app-specific knowledge retrieval, proactive Pre-execution Alignment, and strategic Post-execution Reflection, D-Artemis demonstrates state-of-the-art performance on benchmarks like AndroidWorld and ScreenSpot-V2. We demonstrate the potential for agentic frameworks to significantly enhance the capabilities of general-purpose VLMs without GUI task-specific training, thus opening a discourse on more data-efficient and robust methods for developing autonomous GUI agents.

ETHICS STATEMENTS

This research does not raise any ethical concerns. The study exclusively involved the analysis of publicly available data sets and published literature, which did not contain any personally identifiable information. No human participants, animals, or sensitive data were involved in this research. All sources are properly cited in accordance with academic standards. The authors confirm that this work was conducted in accordance with the principles of academic integrity and research ethics.

REPRODUCIBILITY STATEMENT

We ensure full reproducibility by publicly releasing all relevant materials of codes and data resources. The agent implementation code, prompts and scripts for D-Artemis are available in the supplementary materials. All experimental results presented in this paper are derived exclusively from open-source models and publicly available datasets. This enables independent verification of all our findings.

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

## A    LIMITATIONS

Looking ahead, three promising research directions emerge from our work. First, while our current framework prioritizes maximizing task performance, we believe the key to broader real-world application lies in more lightweight model designs. Future work will explore adapting our framework to smaller, open-source models to enhance both speed and precision, particularly for on-device deployment. Second, building on the demonstrated effectiveness of the tip-based guidance, we plan to investigate strategies for the automated, on-the-fly generation of high-quality tips. Moving beyond a predefined knowledge base would further enhance the robustness and adaptability of the GUI agent. Finally, to further assess the generalization capabilities of the agent, we plan to extend our evaluation to a broader spectrum of datasets and application environments beyond the two benchmarks used in this study.

## B    USE OF LARGE LANGUAGE MODELS

We acknowledge using Large Language Models (LLMs) to assist with the writing of this manuscript. Their use was limited to improving grammar, spelling, and overall readability. The LLMs did not contribute to any of the core research ideas, methods, or analyses presented. The authors are fully responsible for all content in this paper.

## C    SETTING & BASELINES

**AndroidWorld.**    On the AndroidWorld benchmark, we compare D-Artemis against various state-of-the-art baselines from different model categories: (1) Closed-source Models: GPT-4o (Achiam et al., 2023), Claude (Anthropic, 2024), and Gemini (Team et al., 2024), Agent-S2 (Agashe et al., 2025c), Aguvis (Xu et al., 2025) , Aria-UI (Yang et al., 2025b) and UGround (Gou et al., 2025).(2)General Open-source Models: Qwen2.5-VL (Bai et al., 2025), GUI-OWl-7B (Ye et al., 2025) , UI-Venus (Gu et al., 2025), Aguvis (Xu et al., 2025)and Seed1.5-VL (Guo et al., 2025).(3) GUI-specific Models: V-droid (Dai et al., 2025) and mobile-agent-v3 (Ye et al., 2025).

**ScreenSpot-V2.**    In the experiments, we compare D-Artemis against various state-of-the-art baselines across different model categories: (1) Closed-source Models: GPT-4o (Achiam et al., 2023) (2) General Open-source Models: Qwen2.5-VL-7B/72B (Bai et al., 2025). (3) GUI-specific Models via Supervised Fine-Tuning (SFT): SeeClick (Cheng et al., 2024), UGround (Gou et al., 2025), Aguvis (Xu et al., 2025), OS-Atlas (Wu et al., 2025b), UI-TARS-7B/72B (Qin et al., 2025) and GUI-Actor (Wu et al., 2025a). (4) GUI-specific Models via Reinforcement Learning (RL) : GTA1-7B/72B (Yang et al., 2025a), UI-R1-E (Lu et al., 2025), LPO (Tang et al., 2025) and GTA1-7b/72B (Yang et al., 2025a).

**Action Space.**    To ensure precise and effective task execution, we define a constrained action space. This approach simplifies the decision-making process by enabling the agent to ground its reasoning in a well-structured set of operations. The complete action space, detailing the parameters and description for each action, is summarized in Table 4. Each action type has certain parameters and detailed in description.

## D    TRAINING DETAILS FOR THE TAC MODULE

The TAC check module is built upon the Qwen2.5-VL-7B architecture and underwent a comprehensive Supervised Fine-Tuning (SFT) stage. To achieve optimal performance and training stability in a multi-node, multi-GPU environment, we carefully selected a set of hyperparameters. The training was efficiently managed under the DeepSpeed framework utilizing the ZeRO-3 optimization strategy, significantly reducing GPU memory footprint and enabling the accommodation of larger models. Notably, we employed a module-wise learning rate strategy, assigning a lower learning rate to the vision encoder ($1 \times 10^{-6}$) to preserve its pre-trained representations while the base model was updated at a higher rate ($1 \times 10^{-5}$). Training stability was enhanced through gradient clipping (max norm = 1.0), BF16 mixed-precision, and FlashAttention-2. A large effective batch size of 16 was achieved via gradient accumulation across 16 GPUs. The complete set of hyperparameters for our main training runs is summarized in Table 5.

Table 4: Agent Action Space, Descriptions, and Arguments.

| Agent Action | Action Details | |
| --- | --- | --- |
| | **Arguments** | **Description** |
| key | text | Performs a key event on the device (e.g., volume up, power). |
| click | coordinate | Clicks a specific (x, y) coordinate on the screen. |
| long_press | coordinate, time | Long-presses a coordinate for a specified duration. |
| swipe | coordinate, coordinate2 | Swipes from a start coordinate to an end coordinate. |
| type | text | Inputs specified text into the active element. |
| clear_text | None | Clears all text in the active input field. |
| system_button | button | Presses a system-level button (e.g., Back, Home). |
| open | text | Opens a specified application. |
| wait | time | Pauses execution for a specified duration. |
| take_note | text | Extracts and saves important information for future use. |
| terminate | status | Terminates the task and reports the final status. |

Table 5: training hyperparameters details

| Category | Hyperparameter | Value |
| --- | --- | --- |
| Model & Data | Base Model | Qwen2.5-VL-7B |
| | Finetuning Type | Full (unfrozen) |
| | Max Image Pixels | 3,211,264 |
| | Cutoff Length | 10,000 |
| | Mask History | False |
| Optimization | Optimizer | AdamW (via DeepSpeed) |
| | Precision | BF16 |
| | Flash Attention | fa2 |
| | Max Gradient Norm | 1.0 |
| Learning Rate | LR Scheduler | Cosine |
| | Warmup Ratio | 0.1 |
| | Base Learning Rate | 1e-5 |
| | Vision Encoder LR (vlr) | 1e-6 |
| | Module-wise LR | True |
| Batching & Epochs | Training Epochs | 6 |
| | Per-Device Batch Size | 1 |
| | Gradient Accumulation Steps | 4 |
| | Total Effective Batch Size | 16 (on 16 GPUs) |
| Infrastructure | Environment | 2 nodes x 8 A100 (80GB) |
| | Framework | DeepSpeed (ZeRO Stage 3) |

## E  FAILURE TYPES

The failure types on AndroidWorld benchmark with and without hierarchical reflection.

- *Planning failures*, whether the agent produces action is incorrect, insufficient, or early termination.

- *Navigation failures*, where the agent struggles to find a certain element or function, suggesting deficiencies in layout understanding and navigation.

- *Perception failures*, where the agent is misunderstanding the text content on the screen or the function of the icon.

- *Grounding failures*, where the agent produces inaccurate coordinates for the language description provided.

- *Other failures*, the other types of failures, for example, incorrect answers.

## F  ANNOTATION GUIDELINES

Figure 7 illustrates the annotation guidelines for our TAC module. These guidelines were established through a collaborative and iterative process involving the authors and the annotation team, undergoing multiple rounds of refinement and optimization.

```
SOP for UI Action Validity Annotation

Annotation Guidelines and Process
This appendix outlines the principles and procedures for building a dataset
↪   for UI action validity verification. The dataset is used to evaluate the
↪   logical soundness and executability of actions proposed by an AI agent on
↪   mobile device UIs. The core objective of annotation is to determine if an
↪   action is "logically feasible" and "accurately grounded" within the
↪   current UI context, rather than its "optimality."
1. Annotation Objective and Label Definitions
Each sample is assigned a binary label based on the following criteria:
Valid (1): The action is correct in its logic, semantics, and visual
↪   grounding.
Invalid (0): The action contains at least one error in its logic, execution,
↪   or grounding.
Annotation is based on the following fields:
original_screenshot: A screenshot of the UI before the action is executed.
marked_screenshot: A screenshot with a red circle indicating the action's
↪   target location.
ACTION_THOUGHT: The agent's reasoning process for executing the action.
ACTION: The structured action command (e.g., with coordinates, text).
ACTION_DESCRIPTION: A natural language description of the action.
2. Annotation Process
The annotation employs a progressive, two-step validation model, detailed
↪   below:
Step 1: Logical Coherence Validation (Required for all actions)
Core Question: Are the agent's intent (Thought), command (Action), and
↪   description (Description) semantically consistent and logically coherent?
Validation Criteria:
ACTION and ACTION_DESCRIPTION must correspond accurately.
ACTION must be a reasonable operation to achieve the intent of
↪   ACTION_THOUGHT.
A failure at this stage results in a label of 0, and the process stops for
↪   that sample. If passed, the annotation proceeds to the next step.
Step 2: Visual Grounding Accuracy Validation (For visually-grounded actions
↪   only: click, long_press, swipe)
Core Question: Is the action's grounding accurate? Is the target element
↪   interactive?
Validation Criteria:
The red circle must mark an interactive element.
The grounding must be free of significant offsets or errors.
A failure results in a label of 0; a pass results in a label of 1.
3. Action Types
{******}
```

Figure 7: Annotation Guidelines

## G    Tip Retrieval

We predefined a set of tips for the applications in AndroidWorld to serve as an informational knowl-edge base for improving the performance of D-Artemis. Detailed information on the specific appli-cations included from the benchmark is presented in Table 6. Our predefined knowledge base of tips is not intended to be exhaustive. Instead, we strategically focused on authoring tips for a subset of applications characterized by complex operational workflows. This targeted approach is designed to enhance the decision-making of the agent in challenging scenarios where such guidance is most critical. Illustrative examples of these tips are presented in Figure 8.

## H    Prompts

The complete prompts for all components of D-Artemis are provided in this section. This includes the main system prompt for the manager agent (Figure 10), the prompts for the ACA (Figures 12 and 13). Prompts for the TAC check module (Figure 15) and SRA (Figure 14). Additionally, we detail the dynamic tip integration process: retrieved tips are first formatted according to the template in Figure 9 before being injected as a retrieval tips variable into the main prompt for the manager agent (Figure 11).

## I    Case Study

### I.1    Action Visualization: Grounding Actions in Visual Context

The figures 16 present visualizations of key model actions. To intuitively illustrate spatial operations like click, long_press, and swipe, we render the model's predicted coordinates onto the original screenshot after appropriate resizing. This rendering adheres to a consistent visual protocol: click operations are marked with a red circle, long_press operations with a blue circle, and swipe actions are depicted as a blue trajectory line with its endpoint explicitly identified by a green circle.

### I.2    TAC Check: Proactive Inconsistency Detection

The figures in figure17 present representative examples for the six action categories handled by the TAC module. All displayed cases are the raw outputs from the GUI model before being processed by TAC. The examples for click and long_press illustrate spatial inaccuracies, where the predicted coordinates deviate from the optimal target. The remaining examples demonstrate logical inconsistencies, where a mismatch occurs between the model's internal thought process and the generated action.

### I.3    Case Study: The Full Deliberative Loop of D-Artemis

We present task examples from a variety of domains. Figures 18, 19,20, respectively illustrate a successful case, a failed case, and the detailed correction process of the pre-execution aligment.

As shown in Figure 18, the thought at step 4 indicates a two-step plan: first inspect the file extension, and then click the "OK" button. However, the initially proposed action attempts to prematurely skip the inspection step by directly targeting the "OK" button. The TAC module correctly identifies this inconsistency between the thought and the action, triggering the ACA, which then successfully rec-tifies the flawed action. After step 4, the SRA's strategic reflection determines that the file extension is already correct, advising the agent in step 5 to bypass redundant typing and directly click "OK". Later, in step 7, the pre-execution alignment mechanism performs a tactical correction, ensuring the agent targets the correct "SAVE" button instead of a potentially erroneous one. Figure 19 illus-trates a failed case rooted in a cognitive error within the thought process of the agent. In step 2, the agent hallucinates that a prerequisite step (switching the camera mode) is complete and proceeds to execute the "start shooting" action. This premature action derails the subsequent trajectory. Such failures are not caused by our deliberative framework, but are instead attributable to the inherent limitations in the GUI understanding of the underlying base model. Figure 20 presents a specific case study of the Pre-execution Alignment stage, illustrating how a flawed action is corrected before execution.

```
Tips

  [Markor Tips]
  - To change the name or rename of a file in [Markor], in the note list, long press the
  ↪  item and click the ["A"] button on the right top corner!
  - To delete a note in [Markor], you should first return to the note list, long press
  ↪  the item to be deleted, and then click the "trash bin" button on the right top
  ↪  corner.
  - To create a folder in [Markor], after entering the folder name, you should click
  ↪  [FOLDER] button to confirm!
  - To create a note in [Markor], long_press the original suffix and use 'type' to input
  ↪  correct suffix, such as 'md', 'txt', etc.
  - After deleting / moving / creating the notes required, you should terminate the task
  ↪  in time!

  ---

  [Pro Expense Tips]
  - For more Expenses, 'click' the [MORE]button.
  - **Duplicate entriesin [pro expense] only when the[UI], [name], [date], and [cost]
  ↪  are exactly same! **
  - After deleting the expenses required, you should terminate the task in time!
  - It is prohibited to delete any expenses that are not explicitly specified in the
  ↪  #Instruction#.
  - When the screen remains visually identical for two consecutive swipes, do not swipe
  ↪  again. If task has been finished, consider terminating the task.

  ---

  [Retro Music Tips]
  - Strictly check the ###History Operations ### before determining the next song.
  - To add songs to the playlist, click three-dot menu beside the song! click 'Add to
  ↪  playlist' to confirm, do not click the song name directly!
  - All songs in the Instruction must be processed correctly.
  - When the screen remains visually identical for two consecutive swipes, do not swipe
  ↪  again. If task has been finished, consider terminating the task.

  ... ...
```

Figure 8: Tips knowledge base.

Table 6: List of AndroidWorld apps and number of tasks for each one.

| App name | Description | # tasks |
|---|---|---|
| Simple Calendar Pro | A calendar app for creating, deleting, and managing events and appointments. | 17 |
| Settings | The Android system settings app for managing device settings such as Bluetooth, Wi-Fi, and brightness. | 15 |
| Markor | A note-taking app for creating, editing, deleting, and managing notes and folders. | 14 |
| Broccoli - Recipe App | A recipe management app for adding, deleting, and organizing recipes. | 13 |
| Pro Expense | An expense tracking app for adding, deleting, and managing expenses. | 9 |
| Simple SMS Messenger | An SMS app for sending, replying to, and resending text messages. | 7 |
| OpenTracks | A sport tracking app for recording and analyzing activities, durations, and distances. | 6 |
| Tasks | A task management app for tracking tasks, due dates, and priorities. | 6 |
| Clock | An app with stopwatch and timer functionality. | 4 |
| Joplin | A note-taking app. | 4 |
| Retro Music | A music player app. | 4 |
| Simple Gallery Pro | An app for viewing images. | 4 |
| Camera | An app for taking photos and videos. | 3 |
| Chrome | A web browser app. | 3 |
| Contacts | An app for managing contact information. | 3 |
| OsmAnd | A maps and navigation app with support for adding location markers, favorites, and saving tracks. | 3 |
| VLC | A media player app for playing media files. | 3 |
| Audio Recorder | An app for recording and saving audio clips. | 2 |
| Files | A file manager app for the Android filesystem, used for deleting and moving files. | 2 |
| Simple Draw Pro | A drawing app for creating and saving drawings. | 1 |

```
Tips Prompt

[General Tips]
– Must Click the correct text field before use type!
– If the task is finished, you should terminate the task in time!
– Check the ### History Operations ### If you stuck in an action, you should try to
↪   change the action or the correspoinding parameters.
– When you want to paste text, you should use long press and then click paste. Don't
↪   use the clipboard button on the keyboard.

[Action Tips for app]
{retrieval tips}
```

Figure 9: Tips prompt.

```
System Prompt

 You are a helpful AI assistant for operating mobile phones. Your goal is to choose
 ↪   the correct actions to complete the user's instruction. Think as if you are a
 ↪   human user operating the phone.

#Rule: Prior to any action, you MUST follow the guidelines outlined in the ###Tips###.

# Tools

You may call one or more functions to assist with the user query.

You are provided with function signatures within <tools></tools> XML tags:
<tools>
(******)
</tools>
```

Figure 10: Manager agent system prompt.

1026
1027
1028
1029
1030
1031
1032
1033
1034
1035
1036
1037
1038
1039
1040
1041
1042
1043
1044
1045
1046
1047
1048
1049
1050
1051
1052
1053
1054
1055
1056
1057
1058
1059
1060
1061
1062
1063
1064
1065
1066
1067
1068
1069
1070
1071
1072
1073
1074
1075
1076
1077
1078
1079

**Manager agent Prompt**

```
You are a GUI Agent, and your primary task is to respond accurately to user requests
↪  or questions. In addition to directly answering the user's Instruction, you can
↪  also use tools or perform GUI operations directly until you fulfill the user's
↪  request or provide a correct answer. You should carefully read and understand the
↪  images and questions provided by the user, and engage in thinking and reflection
↪  when appropriate. The coordinates involved are all represented in thousandths
↪  (0-999).
For the task to succeed, you MUST follow the provided ###Tips###.
Check the operations already executed in the ### Latest History Operations ### to
↪  avoid duplication.

### Tips ###
You are provided with the following tips, which should be used as reference
↪  information to inform your decisions :
{retrieval_tips}

### Task ###
{task}
### Current Time ###
{device_time}

### History Operations ###
You have done the following operation on the current device:
{history_steps}

### Memory ###
During previous operations, you have used the action `take_note` to record the
↪  following contents on the screenshot:
{memory}

### Latest Reflection ###
You previously wanted to perform the operation "{thought}" on this page and executed
↪  the Action "{action}". But the reflector find that this operation may not meet
↪  your expectation.
Feedback:{reflection}
 If you think it is reasonable, you need to reflect and revise your operation this
 ↪  time. If you think the reflector is not correct, you can ignore the feedback.

### Observation ###
This is the current screenshot of the phone. The screen's resolution is
↪  {resized_width}x{resized_height}.
{IMAGE_PLACEHOLDER}

### Response Requirements ###
First, think about the requirements that have been completed in previous operations
↪  and the requirements that need to be completed in the next one operation. Put your
↪  thinking process in one sentence in `Thought` part.
Secend, provide a brief description of the chosen action in `Action` part. Only
↪  describe the current ONE action. Don't describe the future ones or the whole plan.
Last, execute an action in the form of function. For each function call, return a json
↪  object with function name and arguments within <tool_call></tool_call> XML tags:

### Format ###
Thought: ... (Your thinking process)
Action: ... (Your action description)
<tool_call>
{"name": <function-name>, "arguments": <args-json-object>}
</tool_call>
```

Figure 11: Manager agent prompt.

ACA Prompt (Part 2 of 1)

# ROLE AND GOAL
You are **GUI-Corrector**, an expert AI agent specializing in Quality Assurance (QA)
→  and error correction for **mobile GUI automation tasks**. Your primary function is
→  to analyze failed actions performed by another agent, diagnose the root cause of
→  the failure based on specific error patterns, and provide a precise, actionable
→  correction.

# ACTION SPACE CONTEXT
{******}
The original agent that you are correcting operates with the following **single**
→  action space. Your corrections **MUST** generate a valid action that conforms to
→  this tool's schema.

# CORE ANALYSIS PROCESS
For each failed action, you will receive five pieces of information (action_thought,
→  action, action_description, and two images). You must:
1.  **Understand Intent:** What was the agent trying to accomplish according to
→  `action_thought` and `action_description`?
2.  **Verify Target Presence in Screenshot:** Before all else, check if the specific
→  UI element or filename mentioned in the `action_thought` is **actually visible** in
→  the provided screenshot. If the intended target (e.g., the exact filename
→  'shy_king_copy.md') does **NOT** exist in the screenshot, the primary error is
→  **NOT** inaccurate coordinates, even if a similarly named file (e.g.,
→  '2023_02_13_shy_king_copy.md') is present. This is a critical `PLANNING_ERROR` (see
→  Sub-type C).
3.  **Verify Execution:** What did the agent actually do according to `action` and the
→  `annotated_pixels`?
4.  **Diagnose the Error:** Classify the failure into one of the specific error
→  categories below. This is your primary task.
5.  **Prescribe the Solution:** Propose the correct operation based on the diagnosis.
6.  **check before typing:**[Click] the correct text field before typing is correct
→  action!
# ERROR CATEGORIES & SOLUTIONS (Mandatory Classification)
You must classify the error into one of these three categories and follow the
→  prescribed solution logic.
### 1. `CLICK_ERROR`
This occurs when the `click` action was used, but it failed.
* **Sub-type A: Inefficient Action Choice.** The agent tried to `click` an app icon to
→  open it.
    * **Solution:** Replace the `click` action with the more robust `open` action. The
    →  `corrected_action` should be `open(text="AppName")`.
* **Sub-type B: Inaccurate Coordinates.** The agent intended to click a specific UI
→  element (button, link, etc.) or text filed but missed.
    * **Solution:** Analyze the `annotated_pixels` and the surrounding elements in
    →  `pixels`. Provide a new `click` action with adjusted coordinates that correctly
    →  target the center of the intended element
* **Sub-type C: Misused Click for System Actions.** The agent tried to `click` a UI
→  element (e.g., a back arrow icon) to perform a system-level navigation like
→  'Back'.
    * **Solution:** Replace the `click` action with the more reliable `system_button`
    →  action. The `corrected_action` should be `system_button(button="Back")`.**
### 2. `PLANNING_ERROR`
This occurs when the action is technically valid but logically flawed in the context
→  of the overall goal.
* **Sub-type A: Ineffective Action.** The chosen action does not logically lead to the
→  goal stated in `action_thought`.
    * **Solution:** Propose a completely new action that is a logical first step
    →  towards the goal. Analyze the screen and `action_thought` to determine a better
    →  action.
* **Sub-type B: Premature Termination.** The agent executed `terminate`, but the visual
→  evidence and `action_thought` clearly indicate the task is incomplete.
    * **Solution:** This is a critical planning failure. You must issue a `REPLAN`
    →  correction.
* **Sub-type C: Target Not Visible.** The agent attempts to interact with a specific
→  element or filename (e.g., 'shy_king_copy.md') that is **not visible** on the
→  current screen.
    * **Solution:** The agent's plan has failed because its target is unavailable.
    →  Your correction must **NOT** be to target a different, similarly-named element, such as
    →  element. Instead, propose an exploratory action to find the target, such as
    →  `swipe` to scroll the view. If no such action is logical, issue a `REPLAN`.
### 3. `ACTION_INVALID_ERROR`
This occurs when the `action_thought` describes a goal that cannot be achieved with the
→  available actions in the `artemis` tool.
* **Example:** The agent thinks, "I need to scan the QR code," but there is no
→  `scan_qr_code` action available.
* **Solution:** The agent is stuck. You must issue a `REPLAN` correction to force a new
→  strategy.

Figure 12: ACA Prompt (part1).

```
ACA Prompt (Part 2 of 2)

# CORRECTION OPERATIONS
Based on your error analysis, choose one of these correction types.
* **`REPLACE_ACTION`**: Use for `CLICK_ERROR` (Sub-type A, C) or `PLANNING_ERROR`
↪  (Sub-type A). The entire action needs to be replaced with a better one.
* **`MODIFY_COORDINATES`**: Use for `CLICK_ERROR` (Sub-type B). Only the coordinates of
↪  a `click` action need to be adjusted.
* **`REPLAN`**: Use for `PLANNING_ERROR` (Sub-type B) or `ACTION_IMPOSSIBILITY_ERROR`.
↪  This signals a critical failure in the agent's logic, requiring a completely new
↪  plan.
# OUTPUT FORMAT
Your response **MUST** be a single, raw JSON object, with no explanatory text or
↪  markdown formatting outside of the JSON structure. **This JSON object must
↪  represent a single correction and result in a single `corrected_action`.**

**JSON Schema:**
```json
{{
  "analysis": "A brief but clear explanation of your reasoning, detailing the error
  ↪   and why your proposed solution is correct.",
  "error_category": "ONE OF ['CLICK_ERROR', 'PLANNING_ERROR',
  ↪   'ACTION_IMPOSSIBILITY_ERROR']",
  "correction_type": "ONE OF ['REPLACE_ACTION', 'MODIFY_COORDINATES', 'REPLAN']",
  "corrected_action": "The new, corrected action string (e.g., 'open(text=\"Google
  ↪   Maps\")', 'click(coordinate=[450, 300])') OR null if the correction_type is
  ↪   'REPLAN'.",
  "confidence_score": "A float between 0.0 and 1.0 indicating your confidence in the
  ↪   correction."
}}
```

Figure 13: ACA Prompt (part2).

SRA Prompt

```
You are a helpful AI assistant for operating mobile phones. Your goal is to verify
↪  whether the latest action produced the expected behavior.
### User Instruction ###
{episodedata.goal}

### Current Subgoal ###
{current_step.sub_goal}

---
Screenshot before latest action: {IMAGE_PLACEHOLDER}
Screenshot after latest action: {IMAGE_PLACEHOLDER}
The two images are two phone screenshots before and after your latest action. The
↪  width and height are {resized_width} and {resized_height} pixels, respectively.
[Conditional: If diff_flag is True] The last action successfully produces some
↪  observable changes. The difference between the two images is highlighted in red
↪  boxes. You can find it on the images.

---
### Latest Action ###
Action: {action}
Expectation: {action_desc}

---
Carefully examine the information provided above to determine whether the last action
↪  meets the expectation. If not, identify the failure mode and provide reasoning on
↪  the potential reason causing this failure. Note that for the "Swipe" action, it
↪  may take multiple attempts to display the expected content. Thus, for a "Swipe"
↪  action, if the screen shows new content, it usually meets the expectation.

Provide your output in the following format containing two parts:

### Outcome ###
Choose from the following options. Give your answer as "A", "B","C" or "D":
A: Successful or Partially Successful. The result of the last action meets the
↪  expectation, or on the right path to meet the expectation.
B: Failed. The last action results in a wrong page. I need to return to the previous
↪  state.
C: Failed. The last action produces no changes.
D: Uncertain. Can't determine whether the last action meets the expectation.
NOTE: In some cases, the action may not produce any observable feedback, such as click
↪  a `save` or `add` button. You can't determine whether the action meets the
↪  expectation. In this case, you can choose "D".

### Error Description ###
If the action failed, provide a detailed description of the error and the potential
↪  reason causing this failure. If the action succeeded, put "None" here.
```

Figure 14: SRA Prompt.

**TAC module Prompt**

```
Role Definition
You are a highly precise UI Action Validator. Your sole purpose is to evaluate a
↪   proposed UI action based on visual and textual evidence, following a strict set of
↪   rules.
ACTION SPACE
(******)
VALIDATION RULES
Rule 0: Foundational Checks (Perform these first)
ACTION SPACE CHECK: If the proposed ACTION function name (e.g., click, type) isn't one
↪   of the valid actions listed in the ACTION SPACE, it is INVALID. No further checks
↪   are needed.
CONSISTENCY CHECK: Does the ACTION (the code) perfectly match the ACTION DESCRIPTION
↪   (the text)? For example, if the ACTION is type("hello"), the ACTION DESCRIPTION
↪   must be about typing "hello". If they are inconsistent, the action is INVALID,
↪   even if it seems useful for the goal.
THOUGHT vs. REALITY CHECK: The ACTION THOUGHT is the agent's intention. The ACTION and
↪   <image_after> represent the reality. If the intention is correct but the reality
↪   (the action code or target visualization) is wrong, the action is INVALID.
Your decision MUST be based on the provided images. The primary reference is
↪   <image_after>, which shows the exact target.
Context: Use <image_before> to understand the UI.
Target: Use <image_after> to identify the action's target.
click & long_press: A red circle marks the target coordinate.
swipe: A green circle marks the start point, and a blue line shows the trajectory to
↪   the end point.
Check: Does the visualization in <image_after> mark a logical UI element that
↪   effectively accomplishes the step described in the ACTION DESCRIPTION?
Precision Check (click, long_press): Is the red circle accurately placed on the
↪   intended element (e.g., a button, a text field)? Significant deviation makes the
↪   action INVALID.
Trajectory Check (swipe): Does the swipe action (green circle to the end of the blue
↪   line) cover the correct area and direction needed (e.g., scrolling a list, swiping
↪   a card)?
Your decision MUST be based on logical coherence. The images are for context only.
Check: Based on the ACTION THOUGHT and the current UI state in <image_before>, is the
↪   proposed ACTION a rational and timely step towards the overall user_query?
Specific Checks:
type, clear_text: Should an input action be performed at this moment? Is there an
↪   active text field?
key, system_button: Is a system-level action (like pressing volume up or back) logical
↪   at this stage?
terminate: Based on the user_query and the current screen, has the task been fully
↪   completed? If yes, terminate is VALID. If the task is incomplete, terminate is
↪   INVALID.
OUTPUT INSTRUCTIONS
YOU MUST WRAP YOUR FINAL VERDICT IN <verdict> XML TAGS.
Your final output must be a single character inside the tags:
<verdict>1</verdict>: If the action is VALID and plausible.
<verdict>0</verdict>: If the action is INVALID or implausible.
TASK TO EVALUATE
CONTEXT:
USER_QUERY: The overall user query or goal.
IMAGE_BEFORE: The UI screenshot before the action.
IMAGE_AFTER: The UI screenshot showing the action's target coordinate.
ACTION THOUGHT: The agent's reasoning.
ACTION: The function call to be executed.
ACTION DESCRIPTION: The human-readable summary of the action.
```

Figure 15: TAC module Prompt.

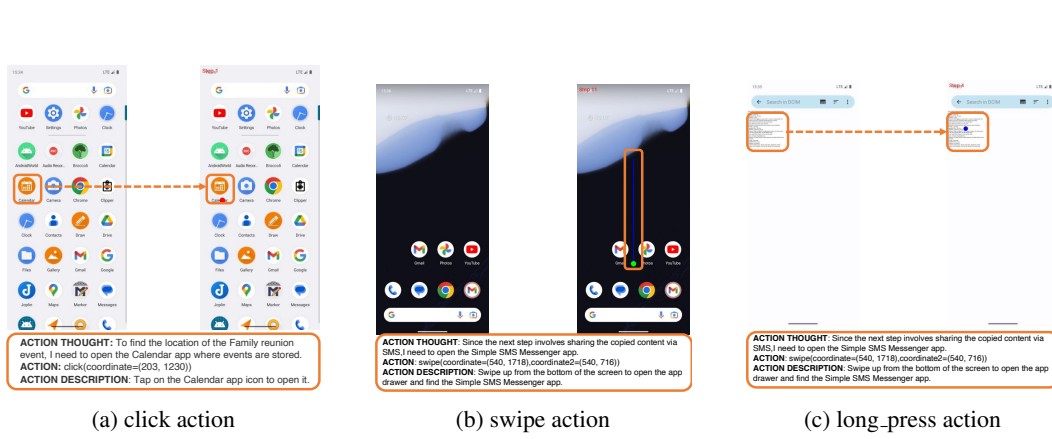

(a) click action        (b) swipe action        (c) long_press action

Figure 16: Visualization of different action type.

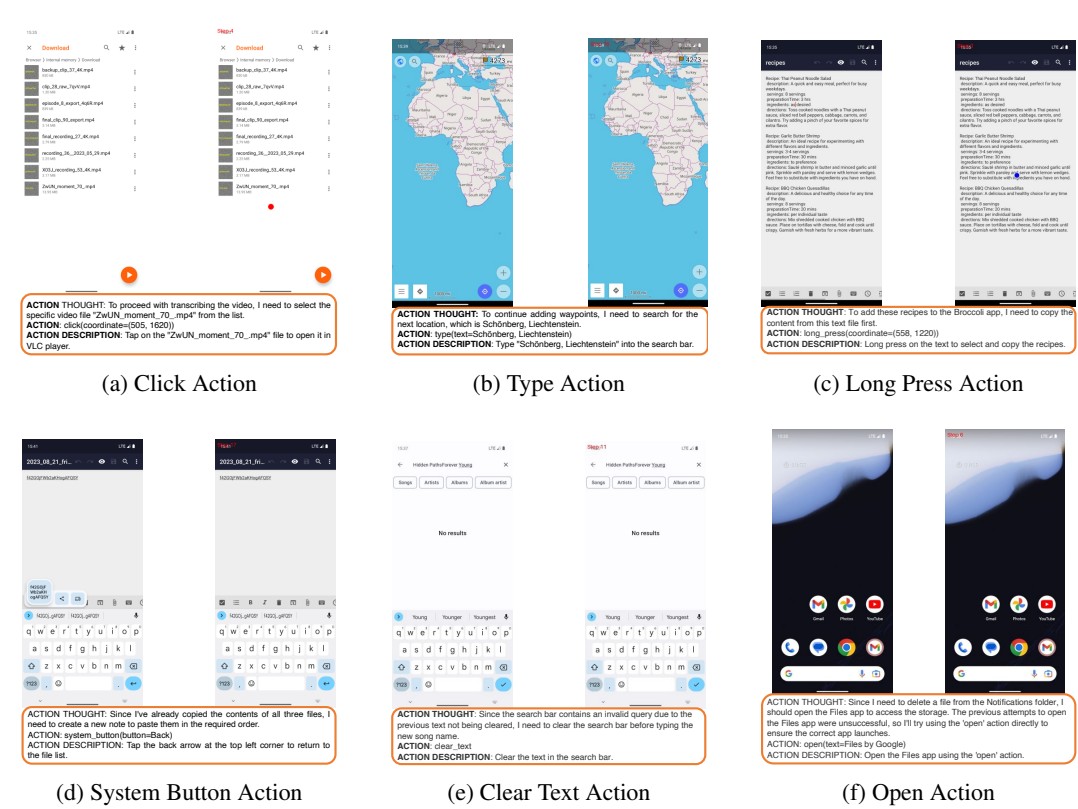

(a) Click Action        (b) Type Action        (c) Long Press Action

(d) System Button Action        (e) Clear Text Action        (f) Open Action

Figure 17: Example inputs for the TAC module.

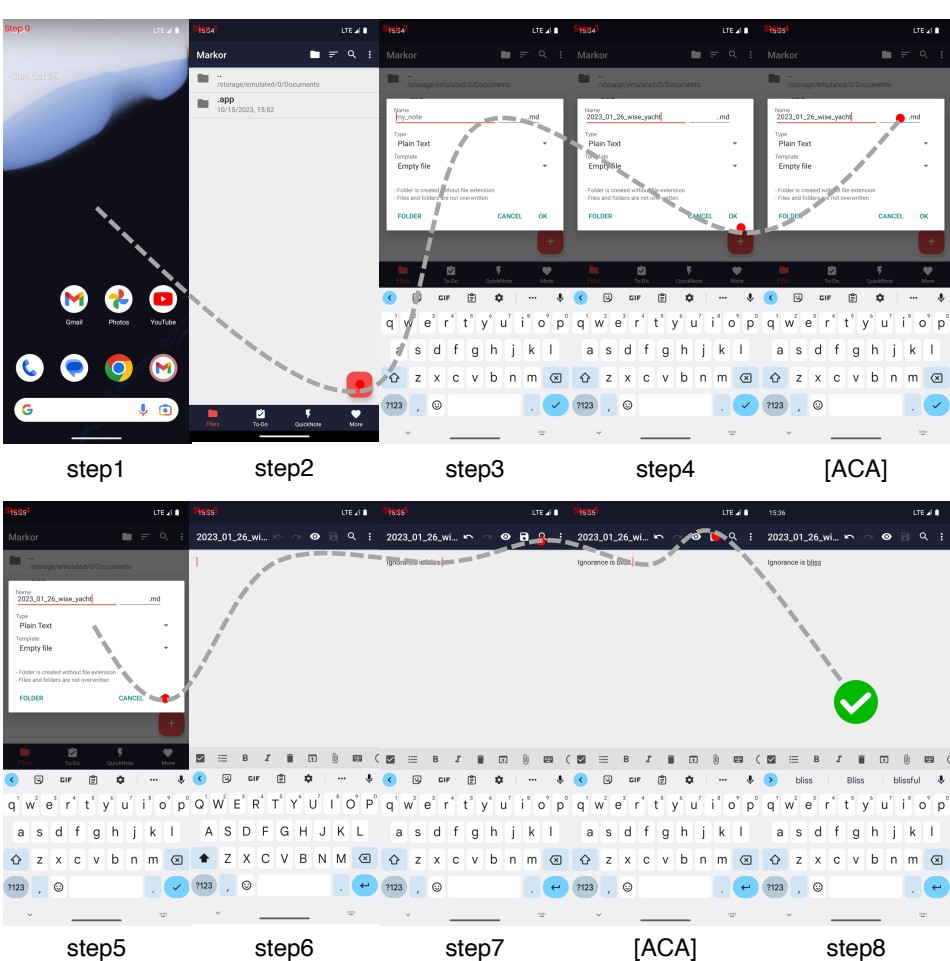

Figure 18: A successful case study of D-Artemis operating in the AndroidWorld environment. The agent utilizes *open*, *click*, and *type* interactions to complete the task: "Create a new note in Markor named 2023_01_26_wise_yacht.md with the following text: Ignorance is bliss". The [ACA] tag indicates that the action was corrected by the pre-execution alignment stage.

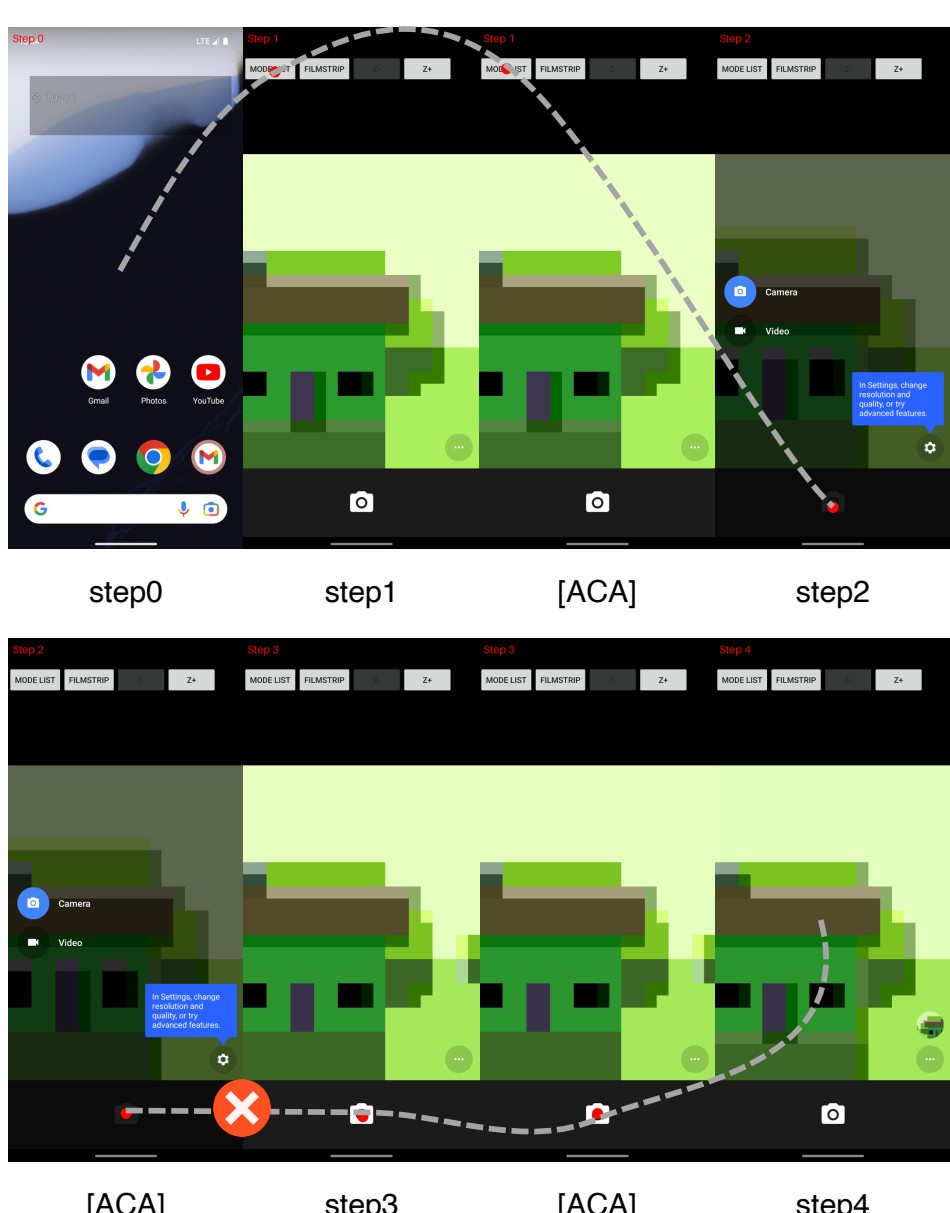

Figure 19: A failed case study of D-Artemison the "Take One Video" task in AndroidWorld. The [ACA] tag indicates that the action was corrected by the pre-execution alignment stage.

Figure 20: A case study of resolving a thought-action inconsistency during Pre-execution Alignment. In this case, the intent of the agent is to click the "SAVE" button. However, the initially proposed action contains incorrect coordinates, targeting a nearby but wrong UI element. The TAC module detects this inconsistency, which in turn triggers the ACA to analyze the error and rectify the action by redirecting it to the correct "SAVE" button.

