# OpenReview forum: "D-Artemis: A Deliberative Cognitive Framework for Mobile GUI Multi-Agents"
_ICLR.cc/2026/Conference — ICLR 2026 Conference Withdrawn Submission_

### Official Review · Reviewer_VMEb · 2025-10-31

**Soundness:** 2
**Presentation:** 3
**Contribution:** 2
**Rating:** 4
**Confidence:** 4

**Summary:**

This paper proposes D-Artemis, a modular cognitive framework that augments GUI agents with pre-execution alignment (TAC + ACA), post-execution reflection (SRA), and app-specific tip retrieval. The framework improves success rates on AndroidWorld (75.8%) and ScreenSpot-V2 (96.8%) without trajectory-based training.

**Strengths:**

- The deliberative architecture is well-motivated and thoughtfully designed.

- Good empirical results +2.5% over prior SOTA on AndroidWorld; meaningful ablations show clear contribution of each module.

- Demonstrates robustness across tasks and solidifies the importance of proactive correction and reflection in GUI agents.

**Weaknesses:**

- The introduced new modules, though well-motivated, are still heuristic and introduce extra model tuning and engineering complexity.

- I didn't find metrics in measuring inference latency or runtime overhead. The system likely increases response time substantially due to extra modules.

- Tip retrieval depends on manually curated, app-specific rules. It's unknown how the system performs without these or on unseen apps.

- Generalization is narrow: Results are limited to two benchmarks. Claims of generalization beyond curated apps are unproven.

**Questions:**

See weaknesses.

---

### Official Review · Reviewer_9Lej · 2025-10-31

**Soundness:** 3
**Presentation:** 3
**Contribution:** 2
**Rating:** 2
**Confidence:** 4

**Summary:**

D-Artemis is a novel deliberative cognitive framework designed to enhance mobile GUI agents by emulating the human cognitive loop of "Thinking, Alignment, and Reflection." It addresses three critical limitations of existing GUI agents: data bottlenecks in end-to-end training, high costs of delayed error detection, and conflicting guidance from generic tips.

The framework includes three main components:
(1) A fine-grained, app-specific tip retrieval to avoid logical conflicts in guidance;
(2) pre-execution alignment: Using TAC and ACA to detect and correct errors before execution
(3) post-execution reflection: Using SRA to enable strategic learning from experience.

The results on both AndroidWorld and ScreenSpot dataset are promising, the ablation study is sufficent.

**Strengths:**

1. This paper proposed a very practical pipeline for real-world GUI analysis and action prediction. The integration of pre-execution alignment and app-specific retrieval is an interesting and effective trick for enhancing performance and reducing prediction failure.

2. The comprehensive evaluation (SOTA comparisons, ablation studies, error analysis) provides strong evidence for the framework’s effectiveness.

3. The paper is well-written.

**Weaknesses:**

1. The three current problems highlighted in the introduction are not fully discussed. Specifically, as for the data bottleneck issue, how your work contributes to this challenge remains unclear based on the current description.

2. The novelty of this paper is not convincing enough. This work presents a solid framework for boosting performance in real-world applications. However, its theoretical and technical advancements are limited and might not be well clarified. It may be better suited for conferences that mainly focus on AI applications.

For example, the TAC and ACA modules are positioned as a core contribution of this work. However, several points require clarification and expansion to strengthen this claim:
First, while the idea of jointly analyzing thoughts and actions pre-execution is intuitive, the TAC module’s scope is limited to coordinate-based actions (e.g., click, swipe), with non-coordinate-based actions excluded from its checks. This narrow focus undermines the module’s broader significance. To justify this design choice, you should explicitly quantify the proportion of coordinate-based vs. non-coordinate-based actions in your full action space.

Second, the methodology section describes the problem setting and pipeline for TAC/ACA, but lacks some theoretical details. Most notably, the "tailored rectification strategy" referenced for the ACA is not demonstrated in this section. You should supplement specific details on how the ACA diagnoses different error types (action type, parameter, invalid action) and applies corresponding corrections—e.g., provide concrete examples of rectification logic for common parameter errors (e.g., incorrect click coordinates) or action type mismatches (e.g., using click instead of long press).


Besides, the description of the Status Reflection Agent (SRA) also lacks clarity. Specifically, the implementation details of the so-called "deep analysis" remain unspecified. While you mention core operations such as "summarization" and "generation," the specific mechanisms for performing these operations—including the criteria or guidelines that govern them to ensure reasonable results and feedback—are not adequately elaborated.

3. Typos or inconsistencies:

The paper is well-written, but there’re still a few typos need to be carefully checked. For instance, line 99 *a effective tip —> an effective tip

"As detailed in Table 2, D-Artemis demonstrates exceptional performance on the ScreenSpot-V2 benchmark. It achieves a 97.9% average success rate, surpassing the previous SOTA, UI-Venus-Ground-72B." However, in table 2, the D-artemis final result is 96.8%.

Overall, while the reported results are promising, the paper’s technical novelty and reproducibility remain the primary concerns in my evaluation.

**Questions:**

1. Could you clarify how your proposed pipeline can solve the data bottleneck problem?

2. Could you clarify some of the missing details mentioned in the weakness session?

3. Could you clarify what the main technical innovations of this proposed pipeline?

4. For practical use, can your pipeline be easily adapted to a totally new app including analyzing the new UI and action according to the given instruction.

---

### Official Review · Reviewer_Cb9V · 2025-11-01

**Soundness:** 3
**Presentation:** 2
**Contribution:** 2
**Rating:** 2
**Confidence:** 4

**Summary:**

This paper proposes a novel multi-agent system for mobile agents. The proposed
framework, D-Artemis, consists of three core modules, Thought-Action
Consistency Check (TAC), Action Correction Agent (ACA), and Status Reflection
Agent (SRA). There are not too much novelty in the framework design, except the
TAC-ACA proactive validation. The experiments show their advantage.

**Strengths:**

1. This paper designs a novel proactive check module based on TAC and ACA, to
  reduce the high cost of delayed error detection and improve the agent
  performance.
2. Adequate improvements are acquired on common GUI benchmarks.

**Weaknesses:**

1. Only one turn of TAC-ACA is performed in the decision loop. How is the
   success rate of one-turn correction? Why aren't multi-loop TAC-ACA cycle
   performed?
2. The results of the actual thought-action inconsistency rate, the detection
   performance (measured by precision & recall) of TAC, the one-turn success
   rate of ACA are missed in the paper.
3. No other multi-agent mobile agent baselines are compared.
4. In Table 3, the number of planning and perception errors of D-Artemis are
   significantly higher than baseline, which lacks an explanation.

**Questions:**

1. How does the name "D-Artemis" come?
2. Aguvis, UI-TARS, MobileUse, and UI-Venus are not "General" models, which is
   labeled mistakenly in Table 1.
3. I suggest include some related work with respect to proactive check or
   imagination-based planning in GUI agents, maybe based on world models.

---

### Official Review · Reviewer_Zy7k · 2025-11-01

**Soundness:** 3
**Presentation:** 3
**Contribution:** 2
**Rating:** 4
**Confidence:** 3

**Summary:**

The paper introduces D-ARTEMIS, a deliberative multi-agent framework for automating mobile GUI tasks. Its core contribution is pre-execution alignment consisting of a Thought-Action Consistency Check module and an Action Correction Agent that validates and corrects actions before execution, rather than after failure. The framework also uses app-specific tip retrieval to avoid logical conflicts and a Status Reflection Agent for post-execution learning.

**Strengths:**

S1.  Pre-execution alignment is intuitive, novel, well-motivated, and addresses a real problem.

S2. Strong result on standard benchmark with proper analysis.

S3. Data Creation process is discussed in detailed with different metrics and IAA.

S4. Idea of Action visualization is clever.

**Weaknesses:**

W1. My major concern is novelty, what authors say as tip guidance, looks similar to RAG with manual instructions. Authors have discussed some of the RAG based approach in related work but I don't see any major difference here. GUI-Critic-R1[1] also proposes pre-operative critic, which looks similar to pre-execution alignment strategy, the main contribution of this paper.

W2. The paper does not properly discuss how well does the method generalise on unseen data or in the wild. Tips knowledge base looks tightly coupled to the AndroidWorld. Authors should discuss more about the generalization or real world demo.

W3. The proposed method inserts additional calls (TAC, ACA, SRA) and uses a fine‑tuned 7B TAC model, but there is no discussion about its impact on token/latency/compute analysis.

---

Minor:

W4. The text at line no 379 claims 97.9% average; Table 2 reports 96.8%.

**Questions:**

Q1.The paper describes the fine-grained, app-specific tip retrieval as its key innovation. However, similar RAG-based retrieval or guideline augmentation[1,2] also provide task or app-specific procedural cues. Similarly for other elements as posed in W1.

Q2. Basically W2. How well the method generalise?

Q3. What are the additional costs compared to baseline? Same as W3.





---

1. Fu, Yao, et al. "Autoguide: Automated generation and selection of context-aware guidelines for large language model agents." Advances in Neural Information Processing Systems 37 (2024): 119919-119948.

2. Kagaya, Tomoyuki, et al. "Rap: Retrieval-augmented planning with contextual memory for multimodal llm agents." arXiv preprint arXiv:2402.03610 (2024).

---

### Note · Authors · 2025-11-21

I have read and agree with the venue's withdrawal policy on behalf of myself and my co-authors.